# Differences between Portuguese and Brazilian Patients with Fibromyalgia Syndrome: Exploring the Associations across Age, Time of Diagnosis, and Fatigue-Related Symptoms

**DOI:** 10.3390/medicina57040322

**Published:** 2021-04-01

**Authors:** Marcos C. Alvarez, Maria Luiza L. Albuquerque, Henrique P. Neiva, Luis Cid, Filipe Rodrigues, Diogo S. Teixeira, Diogo Monteiro

**Affiliations:** 1Department of Sport Sciences, University of Beira Interior, Rua Marquês de Ávila e Bolama, 6201-001 Covilhã, Portugal; marcos.alvarez@ubi.pt (M.C.A.); mluiza.laurentino@gmail.com (M.L.L.A.); hpn@ubi.pt (H.P.N.); 2Research Center in Sport, Health and Human Development (CIDESD), 5000-558 Vila Real, Portugal; luiscid@esdrm.ipsantarem.pt; 3Sport Science School of Rio Maior (ESDRM-IPSantarém), 2040-413 Rio Maior, Portugal; frodrigues@esdrm.ipsantarem.pt; 4Life Quality Research Center (CIEQV), 2040-413 Rio Maior, Portugal; 5Faculty of Physical Education and Sport, Lusófona University (ULHT/FEFD), 376, 1749-024 Lisbon, Portugal; diogo.teixeira@ulusofona.pt; 6Research Center in Sport, Physical Education, and Exercise and Health (CIDEFES), 376, 1749-024 Lisbon, Portugal; 7ESECS—Polytechnique of Leiria, 2411-901 Leiria, Portugal

**Keywords:** fibromyalgia, fatigue, MDF-Fibro-17, cross-cultural comparison

## Abstract

*Background and Objectives*: The current literature demonstrates that different cultures have different perceptions of the symptoms of Fibromyalgia Syndrome (FM). The aim of the study was to explore the differences between Brazilian and Portuguese patients with FM in their fatigue experience and to measure the differences in the perception of fatigue according to age and duration of diagnosis. *Materials and Methods*: In total, 209 Portuguese women aged between 21 and 75 years old (M = 47.44; SD = 10.73) and 429 Brazilian women aged between 18 and 77 years old (M = 46.51; SD = 9.24) were recruited to participate in the present study. Participants filled out the items in the Multidimensional Daily Fatigue-Fibromyalgia-17 Diary (MDF-Fibro-17), a specific tool to measure the level of five components of FM-related fatigue. *Results*: The results showed a greater perception of all of the components of fatigue in the Brazilian sample. No significant differences were found related to the age and duration of FM diagnosis. *Conclusions*: Overall, there are significant differences in fatigue symptoms between Portuguese and Brazilian women with FM, suggesting that cultural and geographical differences should be considered when describing fatigue-related symptoms in women with FM.

## 1. Introduction

Fibromyalgia (FM) syndrome is one of the most common symptoms of chronic musculoskeletal pain and affects mostly the mid-adult population (between 30 and 50 years old), predominately women [1,2]. It is estimated that between 2% and 4% of the world population suffers from FM [3].

Although there is no certainty about the origin of FM, it is currently known that there is a relationship between the nervous system and neurochemical imbalance for the appearance of the first symptoms of this syndrome [4]. The main symptoms reported by patients include musculoskeletal pain without external stimuli, specific places of musculoskeletal sensitivity (also called tender points), decreased muscle strength, chronic fatigue syndrome, and psychological adversities such as sleep disorders, anxiety, and depression [5,6]. FM is considered a multifactorial health condition, that is, its symptoms worsen or are maintained based on the interaction of several biological, psychological, and social factors [7].

Considering the multiple factors that FM involves, the pharmacological treatment for this disease turns out to be generic, only based on the signs and symptoms presented by the patient during the crises [4]. For a doctor to diagnose FM, the patient must achieve high scores in two validated questionnaires by the American College of Rheumatology [8,9]. One of the questionnaires measures pain severity and the other generalized muscle pain intensity [9]. After combining the results of these two questionnaires, and without any change in symptom intensity for more than three months, then the diagnosis of FM is feasible [10]. Fatigue seems to be one of the main symptoms of FM, which patients report to be one of the most disabling harms, thus interfering drastically in their wellbeing and overall quality of life [8,11,12,13]. Recent evidence found that FM patients show less quality of life when compared to people with any other rheumatic disease or syndrome [14,15].

As it stands, fatigue seems to be one of the main disabling symptoms, hindering daily task performance as well as engagement in structured physical exercise [16]. The inability to perform work or day-to-day tasks, not fully understanding the symptoms, feeling of disability, negative impact on interpersonal relationships, isolation, social exclusion, and the difficulty in controlling various symptoms end up leading to a deterioration in the quality of life [17,18]. This causes a significant worsening of health status, both physical and mental, which ends up increasing the severity of the syndrome and its related symptoms in patients with FM [17,18].

### 1.1. Differences between Portugal and Brazil in Fatigue-Related Symptoms

Although similar in language (i.e., Portuguese), Portugal and Brazil seem to display some differences regarding FM and fatigue-related symptoms. In Portugal, the prevalence of the disease is 2.1% of the population, that is, more than 200,000 people have been diagnosed with FM, with the proportion that there are six women with FM for each man with a positive diagnosis [19,20]. A study conducted by Branco et al. [8] showed that Portugal has more patients with FM when compared to other countries in Western Europe, such as Spain, France, Germany, and Italy, which may demonstrate a greater need to explore and examine possible factors and symptoms related to FM. In the economic scope of this syndrome, it is estimated that diseases that have chronic pain-related symptoms (e.g., FM) can cause an average of six annual consultations, generating direct and indirect costs to public health [21].

In Brazil, it is estimated that 2.5% of the population has FM, which represents approximately 4.2 million Brazilians with FM, and the ratio between men and women diagnosed with FM is 1:5.5 [22]. However, in Brazil, due to social disparity and geographical distances from health care centers, patients with FM have more difficulties accessing the necessary care and treatment [21].

In other cross-cultural comparison studies, Kuppens et al. [23] demonstrated that there were significant differences when comparing the perception of fatigue-related symptoms between Belgian and Dutch patients with FM, where Belgian patients reported greater intensity severity in fatigue levels compared to Dutch patients. In this same study, Belgian patients attributed this difference in perception to external factors (i.e., workload, environmental and weather conditions, physical and/or psychological trauma), while Dutch patients attributed fatigue and other FM symptoms to internal factors (i.e., genetic disorders, personality, etc.).

Differences in perceptions of fatigue and FM symptoms were also reported by Ruiz-Montero et al. [24], where it was found that Spanish patients had higher values of perception of FM symptoms compared to patients from Sweden, Belgium, and Holland. Clark et al. [25] compared the perception of FM symptoms between patients in Europe and Latin America, and found that Latin American patients had a greater perception of fatigue and other FM symptoms than European patients. These differences suggest that the perception of fatigue may vary across cultures, even in bordering countries with somewhat similar cultural characteristics (e.g., Belgium and Holland).

### 1.2. Associations between Age, Time of Diagnosis, and Fatigue-Related Symptoms

In the current literature, there is still limited evidence on the effects of fatigue and its relationship with patient’s age and time of diagnosis. Studies have shown that FM-related symptoms are more pronounced in younger patients compared to older patients [26,27,28,29]. These differences in the perception of fatigue between younger and older patients can be explained by lack of experience in controlling this syndrome, which may cause higher levels of symptom intensity when these are reported by younger patients in comparison to older ones [27,30]. Kennedy and Felson [31] carried out a longitudinal study where they monitored the perception of FM symptoms among patients from the day they received the diagnosis and after 10 years. The results showed that in that 10-year period there was a decrease in the scores of fatigue and the general symptoms of FM compared to date of diagnosis. Walitt et al. [32] in their study also used the same research criterion, and 11 years after the first diagnosis, those patients also reported a decrease in the general perception of FM. However, there is a lack of studies regarding the relationship between fatigue-related symptoms, time of diagnosis, and age in FM patients. These studies shed some light on the differences among time of diagnosis and years after dealing with FM.

### 1.3. Current Research

This study aimed to address existing limitations and provide an important incremental step forward regarding how Portuguese and Brazilian women patients with FM experience fatigue-related symptoms, and if there could be cultural differences between them. Thus, the aim of the study was to explore the differences between fatigue-related symptoms in Brazilian and Portuguese women with FM syndrome. The second aim was to examine associations between age, time of diagnosis, and fatigue-related symptoms in both countries.

## 2. Materials and Methods

### 2.1. Participants

Data from two independent samples were collected for the present study. The first sample (Sample 1) comprised data from 290 Portuguese women aged between 21 and 75 years (M = 47.44 ± 10.73) who were invited to participate in the present study. Mean time diagnosis for FM ranged between 1 and 30 years (M = 7.71 ± 6.04), referred by rheumatologists following the suggested patterns for diagnosis of this syndrome. The second sample (Sample 2) comprised data from 429 Brazilian women aged between 18 and 77 years old (M = 46.52; SD = 9.23). Mean time diagnosis for FM ranged between 1 and 30 years (M = 7.86 ± 6.48). The participants of this research were diagnosed with FM by licensed rheumatologists and their diagnosis was established by the recommended guidelines proposed by the American College of Rheumatology [9].

### 2.2. Data Collection Procedures

Before data collection, ethical approval was obtained from the ethical and scientific committee of the Center for Research in Sport, Health and Human Development (CIDESD) under the registration number UID04045/2020. Researchers contacted several hospital boards and health care centers (*n* = 10) to obtain permission to conduct the present research. At this stage, the objectives of the study were explained and endorsement from each board was obtained. Next, rheumatologists (*n* = 8) administered the questionnaire to patients with FM, according to the previously mentioned inclusion criteria. All individuals participated voluntarily in this study, receiving no monetary reward for their contribution and were informed about the study objectives and gave informed consent before participating. The current study was conducted in accordance with the Helsinki declaration and later amendments.

### 2.3. Instrument

The Multidimensional Daily Diary of Fatigue-Fibromyalgia-17 items (MDF-fibro-17) validated for Brazilian and Portuguese populations was used to measure fatigue-related symptoms of FM [33]. The 17 items consist of five subscales: global fatigue experience (four items; item example, “*How severe was your fatigue today?*”); physical fatigue (three items; item example, “*How weak were your muscles today?*”); cognitive fatigue (four items; item example, “*How hard was it to concentrate because you were tired today?*”); motivation (three items; item example, “*How much effort was made today?*”); and impact on function (three items; item example, “*Did you do things more slowly because you were tired today?*”). Participants responded to each item using a 10-point scale, ranging from 0 (“nothing”) to 10 (“extremely”). Higher scores indicated greater fatigue severity. Previous studies support the validity and reliability of this questionnaire [7].

### 2.4. Statistical Analysis

Data were initially exported to IBM SPSS STATISTICS v.23 (IBM Corp., Armonk, NY, USA) for preliminary analyses. Descriptive statistics, including mean, standard deviation, skewness, and kurtosis were calculated for all studied variables in both samples. Cutoffs for normality were determined based on traditional guidelines, accepting scores within −2/+2 and −7/+7 for skewness and kurtosis, respectively. Inspection of missing values and outliers was also carried out.

Subsequently, a t-test for the independent sample was used to analyze differences between fatigue-related symptoms in Brazilian and Portuguese female patients with FM, as well as between time of diagnosis. The Cohen’s d was calculated to obtain the correspondent effect size for t-test analysis, considered as: trivial (0.00–0.19); small (0.20–0.49), average (0.50–0.79), and large (greater than or equal to 0.80).

In the following step, Pearson bivariate correlations were calculated between age, time of diagnosis, and fatigue-related symptoms in each sample. A *p*-value of less than 0.05 was considered significant.

## 3. Results

### 3.1. Preliminary Analysis and Comparison of Fatigue Components between Brazil and Portugal

Inspection of the data showed no missing values and outliers. The descriptive statistics of all fatigue-related symptoms are summarized in Table 1. Examining mean values, Brazilian women reported higher scores in all fatigue-related symptoms compared to Portuguese women. Results revealed no violations of the univariate distribution since skewness and kurtosis were contained between −2 and +2, and −7 and +7, respectively.

The analysis of the t-test for independent samples showed that there were significant differences in all fatigue-related symptoms as a function of country (*p* < 0.01). Effect sizes were small, ranging between 0.28 and 0.45. For detailed information see Table 1.

### 3.2. Associations between Age, Time of Diagnosis, and Fatigue-Related Symptoms

Correlations between age, time of diagnosis, and fatigue-related symptoms in both countries are reported in Table 2. Global experience, physical fatigue, cognitive fatigue, motivation, and impact on function were not significantly correlated with age. Similarly, fatigue-related symptoms were not significantly correlated with time of diagnosis in the Brazilian and Portuguese female patients with FM. Details are displayed in Table 2.

## 4. Discussion

The present research focused on the study of fatigue as one of the main symptoms in Brazilian and Portuguese women with FM. The MDF-Fibro-17 validated for the Brazilian and Portuguese population was used to measure the level of five components of FM-related fatigue symptoms in a large sample of female patients from Brazil (*n* = 429) and Portugal (*n* = 290). The aim of the study was to explore the differences between Brazilian and Portuguese patients in their experience of fatigue. Additionally, associations between fatigue, age, and duration of the diagnosis were carried out in both samples under analysis. Results showed greater perception of all the components of fatigue in the Brazilian sample. No other significant differences were found related to the age of the patients and the duration of FM diagnosis. Implications derived from this study in relationship to fatigue in FM will be discussed according to existing literature.

### 4.1. Comparison of Fatigue Components between Brazil and Portugal

The present results suggest that Brazilian women with FM, when compared to Portuguese women, tend to have more pronounced perception of fatigue-related symptoms compared to Portuguese patients. This finding is consistent with previous studies (e.g., Clark et al. [25]), where European FM patients displayed significant differences in fatigue-related symptoms. The study by Ruiz-Montero et al. [34] found that Spanish women had higher perceptions of FM-related syndrome and higher levels of fatigue-related symptoms than Dutch women. This same panorama was verified in studies by Liedberg, Burckhardt, and Henriksson [35], where different cultures have different perceptions of FM and levels of fatigue from each other. Kuppens et al. [23] in their study made a comparison of the perception of FM symptoms in Belgians and Dutch, and found that Belgian patients had higher perceptions of the symptoms; that is, they more felt the effects caused by fatigue and the other symptoms of FM. Liedberg, Burckhardt, and Henriksson [35], on the other hand, carried out a comparative study between patients from Sweden and the United States with FM, where they demonstrated that North American patients felt the effects of fatigue and other symptoms of FM more than Swedish ones. Thus, it seems that European countries, compared to American countries, tend to perceive fatigue-related symptoms less.

These cultural differences in the perception of FM fatigue may suggest that these different levels of perception may be due to external reasons or even to local influences (that is, differences in the health system, sociodemographic factors, stress levels, weather conditions). These external reasons seem to be related to the differences in symptoms related to fatigue between Brazilian and Portuguese women with FM. Brazil has a larger geographic proportion, making access to health care more difficult. In addition, Brazil ends up having different social realities between states and regions compared to Portugal, which ends up generating social inequalities, thus delaying the diagnosis of FM syndrome and its pharmacological and non-pharmacological treatments [22].

### 4.2. Associations between Age and Fatigue-Related Symptoms of Brazilian and Portuguese Patients with FM

The present study showed that fatigue components were not significantly associated with age, regardless of country as demonstrated by the values presented in Table 2 (*p* > 0.05). This suggests that fatigue does not vary significantly over time regarding FM patients’ perceptions. This is in line with Salinem et al. [35], who demonstrated that fatigue is a longitudinal experience that is maintained throughout the life of patients with FM, without significant changes.

However, the results of this study differ from some studies in the literature, which found that there are differences in the perception of fatigue among younger patients compared to older patients [26,28,29]. The study by Shillam, Jones, and Miller [26] made a comparison between the perception of FM symptoms between middle-aged adult patients and older patients, and the younger ones were found to be more symptomatic, that is, they had greater symptom intensity when compared to older patients. Keskindag [28] in his study compared the perception of fatigue between three age groups (i.e., group 1 = 30–39-year-old patients; group 2 = 40–49-year-old patients; and group 3 = 50–59-year-old patients), and the authors concluded that younger patients (group 1) had higher levels of fatigue when compared to adults in the other groups. Similar findings to this were found by Jiao et al. [29] when comparing fatigue-related symptoms between different age groups. These authors found that after aggregating patients into age groups: group 1 = 18–39 years; group 2 = 40–59 years; and group 3 = > 60 years, younger patients (group 1) had higher levels of fatigue and other symptoms compared to FM patients in groups 2 and 3.

### 4.3. Associations between Time of Diagnosis and Fatigue-Related Symptoms of Brazilian and Portuguese Patients with FM

When analyzing the relationship between fatigue domains and time of diagnosis, results showed that there were no significant differences in both Brazilian and Portuguese samples of women with FM. The results show that the time of diagnosis is not related to the symptoms of fatigue. This result differs from the studies by Kennedy and Felson [31], Walitt et al. [32], and White et al. [36], who followed women newly diagnosed with FM for 3, 10, and 11 years to determine whether the time of diagnosis had an effect on the perception of symptoms of this syndrome. In the study by Walitt et al. [32], 11 years after receiving the diagnosis, it was found that these patients showed lower perceptions of fatigue and other symptoms of FM; in other words, these patients began to report a decrease in the intensity of these symptoms. These results were also verified in the study by Kennedy and Felson [31], where after 10 years following diagnosis, these patients also reported decreased levels of fatigue. The study by White et al. [36] also monitored patients newly diagnosed with FM, and found that three years after receiving diagnosis, these patients reported lower values of fatigue and symptoms of FM.

### 4.4. Limitations

Present research has some limitations that should be acknowledged. First, this study had a cross-sectional design and longitudinal/experimental studies can be helpful in measuring temporal/longitudinal variance of fatigue-related symptoms in both countries. Another limitation of the present study that should be acknowledged is that we did not measure pain severity and intensity. As previously mentioned, fatigue does not occur in isolation; rather, it is concurrently present and varies in severity with other symptoms such as chronic pain, sleep disorders, and other cognitive difficulties. Since both are important in the diagnosis of FM, future studies should examine pain and fatigue-related symptoms of FM and explore possible associations between them. Third, the findings of this study cannot be used for comparison between other cultures, as more research is needed to establish this comparative pattern. Therefore, future studies should explore fatigue-related symptoms across more cultures and compare results with current evidence.

## 5. Conclusions

The uniqueness of this study is the emphasis on the proper use of a valid and invariant measure of fatigue-related symptoms in patients with FM. This was the first study that carried out this type of comparison between these two countries. Overall, there are significant differences in fatigue symptoms between Portuguese and Brazilian women with FM, suggesting that cultural and geographical differences should be considered when describing fatigue-related symptoms in women with FM. Further in-depth studies are needed to assess these factors specifically. In addition, the present study showed that age and time of diagnosis are not associated with fatigue in Portuguese and Brazilian women with FM. The analysis of fatigue components between Portuguese and Brazilians brings new contributions to the literature in the scope of FM, where it reinforces the importance of the robust evaluation of difference in perception of fatigue between different cultures. In general, the findings of this study provide new literature involving the examination of fatigue and may assist future research involving cross-cultural investigations considering patients with FM.

## Figures and Tables

**Table 1 medicina-57-00322-t001:** Descriptive statistics of each variable according to samples (Portugal and Brazil).

		N	M	SD	S	K	t	df	*p*-Value	*d*
Global experience	Portugal	290	7.25	1.58	−0.75	0.49	−3.690	717	<0.001	0.28
	Brazil	429	7.72	1.74	−0.97	0.70				
Physical fatigue	Portugal	290	7.50	1.64	−0.95	1.18	−5.898	717	<0.001	0.44
	Brazil	429	8.22	1.59	−1.02	0.67				
Cognitive fatigue	Portugal	290	7.20	1.86	−0.90	0.89	−4.848	717	<0.001	0.37
	Brazil	429	7.88	1.82	−1.32	1.99				
Motivation	Portugal	290	7.50	1.70	−0.89	0.73	−5.312	717	<0.001	0.40
	Brazil	429	8.19	1.71	−1.18	1.10				
Impact on function	Portugal	290	7.45	1.83	−1.15	1.20	−5.847	717	<0.001	0.45
	Brazil	429	8.27	1.80	−1.67	2.45				

Note. N = sample size; M = mean; SD = standard deviation; S = skewness; k = kurtosis; d = effect size.

**Table 2 medicina-57-00322-t002:** Correlation between age, time of diagnosis, and fatigue in Portugal and Brazil.

Variables	Portugal		Brazil	
	Age	Time of Diagnosis	Age	Time of Diagnosis
Global experience	0.03	0.00	−0.04	0.01
Physical fatigue	0.06	0.02	−0.01	0.03
Cognitive fatigue	0.04	−0.01	−0.04	0.02
Motivation	0.06	0.01	−0.04	−0.01
Impact on function	0.06	0.05	−0.04	0.01

## Data Availability

Due to issues of participant consent, data will not be shared publicly. Interested researchers may contact the corresponding author (diogo.monteiro@ipleiria.pt).

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
