# Peer review of "Differences between Portuguese and Brazilian Patients with Fibromyalgia Syndrome: Exploring the Associations across Age, Time of Diagnosis, and Fatigue-Related Symptoms"

_medicina, 2021, doi:10.3390/medicina57040322_

Round 1

Reviewer 1 Report

  • It could be interesting to comparative the pain with fatigue. In different part of this study, you mix both symptoms. Do you think if the pain could be influence to fatigue?.
  • Under what criteria has the diagnostic of the patients been established?. The criteria of fibromyalgia have been modified over the years. 
  • Diagnostic time: could be interesting to reclute when the patients began to felt their symptoms. This time could be biased by accessing to medical services. Based on what criteria was the mean calculated?.
  • In the line number 263 to 269, you comparative the pain with fatigue. I think that it cannot be compared. 

Author Response

Reviewer: 1

General comments:

It could be interesting to comparative the pain with fatigue. In different part of this study, you mix both symptoms. Do you think if the pain could be influence to fatigue?

R: We appreciate the suggestion. Unfortunately, literature is scarce on the association between pain and fatigue (Nicassio et al.2002). In this study, we aimed to compare mean scores of fatigue symptoms between two groups of women from different countries.

Under what criteria has the diagnostic of the patient's been established? The criteria of fibromyalgia have been modified over the years.

R: Interesting point. The patients of this research were diagnosed with fibromyalgia by licensed rheumatologists, and their diagnosis was established by what is recommended by the American College of Rheumatology. This information was added in the manuscript (Page 3, Line 110).

Diagnostic time: could be interesting to reclute when the patients began to felt their symptoms. This time could be biased by accessing to medical services. Based on what criteria was the mean calculated?

R: We apologize but are unable to understand your comment in full. Nonetheless, it is worth to mention that the mean time for diagnosis was calculated based on electronic logs from the medical archives. Data was provided by the rheumatologist.

In the line number 263 to 269, you comparative the pain with fatigue. I think that it cannot be compared.

R: We apologize for this typo. This section was revised (Page 6, Line 218 to 225)

Reviewer 2 Report

The present research is focused on the study of fatigue as one of the main symptoms in fibromyalgia syndrome (FM). The authors use a specific tool, the Multidimensional Daily Diary of Fatigue-Fibromyalgia-17 items (MDF-Fibro-17), to measure the level of five components of FM-related fatigue in a large sample of patients from Brazil (429) and Portugal (290). The aim of the study is to explore the differences between Brazilian and Portuguese patients in their experience of fatigue and to confirm the presence of changes in the perception of fatigue as a function of the age of patients or the duration of the diagnosis. Results showed greater perception of all the components of fatigue in the Brazilian group. No other significant differences have been found in relation to the age of the patients and the duration of FM diagnosis. The authors discuss about the implications derived from this study in relation to physical activity, quality of life, persistence and impact of fatigue in FM.

The core of this study is interesting because the fatigue is a really prominent symptom of FM but the research on this concern is scarce. Also, the large sample size of participants recruited from two different countries is a valuable point of this research paper. However, in my opinion, the data extracted from these strong points have not been totally exploited and important key aspects should be improved. 

First of all, in the "Introduction" section, FM syndrome is adequately described but fatigue is introduced as “the main symptom” of FM (line 62) and this is not correct. Fatigue is an important symptom, one of the main symptoms and necessary for the diagnosis, but the principal symptom in FM is generalized widespread pain. That’s why I think a measure of pain is essential in the study of FM. The impact of FM in quality of life is highlighted in the introduction, nevertheless no measure of this variable is included in the study. In spite of this lack, the objective is clear and modest: to explore the fatigue components in FM patients. This idea is well stablished and argued in the theoretical background, but more bibliographic references supporting the influence of “age” and “time of diagnosis” in the evolution of fatigue (or other symptoms in FM) would be welcomed. In addition, the presence of explicit hypothesis in accordance with the reviewed literature is recommended.

Regarding the “Materials and methods” section, the procedure is well described, in order to assure the replicability of the study. Some information should be rewritten for a better understanding: “The Portuguese participants were diagnosed with FM in a physician who was 7.71 ± 6.04 years old” (line 84-85), it seems you are describing the age of the physician. In the same way “Brazilian women were diagnosis with FM on average 7.86 ± 6.48 years” is an incomplete sentence. The English writing should be checked. 

Two group based on “diagnosis time” were created, taking into account the “media value” of each group of participants. Given that the mean time of diagnosis is 7.71 ± 6.04 years for Portuguese and 7.86 ± 6.48 years for Brazilian, I guess the cut-off points of 5 and 6 respectively are referred to “median value”. In addition, 3 “age groups” where stablished for each groups of participants: 18-29, 30-50 and >50 years. Regardless of this method is not wrong and it is according with the selected statistical analyses, other methods would be more appropriated to test the presented objectives (see below).

T-tests for independent samples were performed to explore the differences between Brazilian and Portuguese groups on the score of the five components of fatigue. I consider this is a good initial step of the statistical analysis to test the possible differences on fatigue between groups. Going further, multivariate analysis of variance (MANOVA) would be a better way to explore these data, introducing cultural group (Brazilian and Portuguese) as fixed factor and the fatigue measures as dependent variables.

In the next step, the collected data are divided in categories. One way ANOVAs were performed on three groups of age (per each group of participants). The cut-offs are stablished ad-hoc and the G1 of age is smaller than the others (17 – 158 – 115 in the Portuguese group, 12 – 267 – 150 in the Brazilian group). To obtain reliably results, it would be recommended similar sample size groups, or at least with an adequate size representing the population. As the G1 doesn’t reach this condition and the involved variables are continuous (age and score in fatigue components), I suggest carrying out bivariate Spearman correlation to explore the possible association between these two variables. Or even linear regression analysis could be used to define the predictive role of age on fatigue scores. In the same way, comparisons between categories based on the time of diagnosis, could be reanalyzed with Spearmen correlation or linear regression analysis, instead T-student test. In this case, the classification on two groups above and below the median value might not allow to discriminate the possible differences between the cases with a value near to the median.  

The “Results” section shows direct and concise information, but it would be appreciated to read the main result for each analysis, as you do in lines 138-139, not only the indication to see the corresponding table (e.g., One way ANOVA showed no significant effect (p>0.05) on fatigue among the different age groups). The results could be presented in fewer number of tables, so the data in Tables 1 and 2 could be put together, as well as Tables 5 and 6, 7 and 8.

The “Discussion” section need a deep review because there are numerous speculative statements, without any support in the reported findings. Here there are some examples:

Line 253 “This demonstrates that physical activity is beneficial for pain control…” No, the study doesn’t include a measure of physical activity, so it can´t be demonstrated.

Line 265 “This demonstrated that fatigue, and its components, tend to be relevant symptoms and that they deteriorate the quality of life of FM patients, regardless of their age.” No, the data showed that fatigue scores are similar between different age ranges. There is not a measure of other symptoms of FM or quality of life, so it can’t be demonstrated by the present results.

Line 268 “This may reveal that fatigue is the most persistent symptom and that there is no slowing down over the years in FM patients.” There is not a measure of other symptoms of FM for the comparison with fatigue, so it can’t be revealed by the present results.

Line 295 “The present study reinforces the real importance of assessing fatigue in patients with FM, as it is an extremely relevant symptom”. The present results showed a significant difference in fatigue scores between Portuguese and Brazilian FM patients, that’s all. Such statement can’t be declared on the basis of the present results.

It is important to be cautious and modest when interpreting the results. Even more when the data collected and the procedure are easy and direct, like in the present study. I suggest to improve the statistical analyses, think and pay attention to the interpretation of the results and rewrite with caution the discussion and conclusions. Additionally, it would be great to check the English language in the whole manuscript and complete the introduction with more bibliographic resources supporting your objective and hypothesis. Results also could be completed with the description of the obtained results.

Author Response

Reviewer: 2

The present research is focused on the study of fatigue as one of the main symptoms in fibromyalgia syndrome (FM). The authors use a specific tool, the Multidimensional Daily Diary of Fatigue-Fibromyalgia-17 items (MDF-Fibro-17), to measure the level of five components of FM-related fatigue in a large sample of patients from Brazil (429) and Portugal (290). The aim of the study is to explore the differences between Brazilian and Portuguese patients in their experience of fatigue and to confirm the presence of changes in the perception of fatigue as a function of the age of patients or the duration of the diagnosis. Results showed greater perception of all the components of fatigue in the Brazilian group. No other significant differences have been found in relation to the age of the patients and the duration of FM diagnosis. The authors discuss about the implications derived from this study in relation to physical activity, quality of life, persistence and impact of fatigue in FM.

The core of this study is interesting because the fatigue is a really prominent symptom of FM but the research on this concern is scarce. Also, the large sample size of participants recruited from two different countries is a valuable point of this research paper. However, in my opinion, the data extracted from these strong points have not been totally exploited and important key aspects should be improved. 

R: We appreciate your time in this review process. Substantial revisions were made. Changes are marked using the track-change option in MS Word. Page and line numbers are also marked for aiding the reviewers in the review process.

First of all, in the "Introduction" section, FM syndrome is adequately described but fatigue is introduced as “the main symptom” of FM (line 62) and this is not correct. Fatigue is an important symptom, one of the main symptoms and necessary for the diagnosis, but the principal symptom in FM is generalized widespread pain.

R: We apologize for considering fatigue as the main symptom. It is one of the most prevalent symptoms, as well as others (e.g., pain, muscle stiffness, sleep disorder). As it stands, fatigue seems to be one of the main disabling symptoms, hindering daily task performance as well as engaging in structured physical exercise. Sentences were revised to a more conservative manner (Page 2, Line 51).

That’s why I think a measure of pain is essential in the study of FM.

R: We appreciate your suggestion and agree that future studies should measure both pain and fatigue-related symptoms of FM. Limitation and future research suggestion was added in the discussion section (Page 6, Line 238 to 246).

The impact of FM in quality of life is highlighted in the introduction, nevertheless no measure of this variable is included in the study.

R: The reviewer is right. We did not measured quality of life as it was not the main goal of the present research. We removed sentences that gave the impression of the measurement of quality of life, stating only possible associations with fatigue in patients with FM.

In spite of this lack, the objective is clear and modest: to explore the fatigue components in FM patients. This idea is well stablished and argued in the theoretical background, but more bibliographic references supporting the influence of “age” and “time of diagnosis” in the evolution of fatigue (or other symptoms in FM) would be welcomed.

R:  This section was substantially revised and more literature was added justifying the need to examine possible associations between age, time of diagnosis, and fatigue-related symptoms across countries (Page 2 and 3, Line 62 to 95).

Regarding the “Materials and methods” section, the procedure is well described, in order to assure the replicability of the study. Some information should be rewritten for a better understanding: “The Portuguese participants were diagnosed with FM in a physician who was 7.71 ± 6.04 years old” (line 84-85), it seems you are describing the age of the physician. In the same way “Brazilian women were diagnosis with FM on average 7.86 ± 6.48 years” is an incomplete sentence. The English writing should be checked.

R: The entire manuscript was proof-read by an English teacher. The Material and method section was substantially revised (Page 3, Line 101).

Two group based on “diagnosis time” were created, taking into account the “media value” of each group of participants. Given that the mean time of diagnosis is 7.71 ± 6.04 years for Portuguese and 7.86 ± 6.48 years for Brazilian, I guess the cut-off points of 5 and 6 respectively are referred to “median value”. In addition, 3 “age groups” where stablished for each groups of participants: 18-29, 30-50 and >50 years. Regardless of this method is not wrong and it is according with the selected statistical analyses, other methods would be more appropriated to test the presented objectives (see below).

R: Significant changes were made in the statistical analyses section (Page 4, Line 132).

T-tests for independent samples were performed to explore the differences between Brazilian and Portuguese groups on the score of the five components of fatigue. I consider this is a good initial step of the statistical analysis to test the possible differences on fatigue between groups. Going further, multivariate analysis of variance (MANOVA) would be a better way to explore these data, introducing cultural group (Brazilian and Portuguese) as fixed factor and the fatigue measures as dependent variables.

R: We thank the reviewer for his/her suggestion on conducting MANOVA. We calculated the variance to explore differences between countries, introducing cultural group (Brazilian and Portuguese) as fixed factor and the fatigue measures as dependent variables. The results did not add much information beyond what was observed in the t-test analysis. First, Lambda Wilks was significant (Lambda = .939; p<0.001, eta square .061). Second, differences in fatigue-related symptoms between countries were all significant (p<0.001). We believe that adding this information would confuse the readers. However, Cohen’s d was added as a mean to examine effect size across variables under analysis.

In the next step, the collected data are divided in categories. One way ANOVAs were performed on three groups of age (per each group of participants). The cut-offs are stablished ad-hoc and the G1 of age is smaller than the others (17 – 158 – 115 in the Portuguese group, 12 – 267 – 150 in the Brazilian group). To obtain reliably results, it would be recommended similar sample size groups, or at least with an adequate size representing the population. As the G1 doesn’t reach this condition and the involved variables are continuous (age and score in fatigue components), I suggest carrying out bivariate Spearman correlation to explore the possible association between these two variables. Or even linear regression analysis could be used to define the predictive role of age on fatigue scores. In the same way, comparisons between categories based on the time of diagnosis, could be reanalyzed with Spearmen correlation or linear regression analysis, instead T-student test. In this case, the classification on two groups above and below the median value might not allow to discriminate the possible differences between the cases with a value near to the median. 

R: We appreciate your crucial comments. We agree that G1 group did not meet the criteria for statistical sample power. Age and time of diagnosis were treated as continuous variables in both countries. First, bivariate correlations were calculated for possible associations among these variables and fatigue-related symptoms across countries. Results displayed no significant associations (see table 2). Thus, linear regressions were not carried out. As it stands, results are quite similar to those reported in the first manuscript: age and time of diagnosis are not associated with fatigue-related symptoms. These are still interesting results, as they provide evidence on how interventions should be carried out on patients with FM, and how age and time of treatment do not affect fatigue. Please consider the revised statistical analysis, as well as the results and discussion sections on this topic (Page 4, Line 132).

The “Results” section shows direct and concise information, but it would be appreciated to read the main result for each analysis, as you do in lines 138-139, not only the indication to see the corresponding table (e.g., One way ANOVA showed no significant effect (p>0.05) on fatigue among the different age groups). The results could be presented in fewer number of tables, so the data in Tables 1 and 2 could be put together, as well as Tables 5 and 6, 7 and 8.

R: We followed your advice and merged tables for clarity (see Table 1). Please consider the revised result section for your review.

The “Discussion” section need a deep review because there are numerous speculative statements, without any support in the reported findings. Here there are some examples:

R: The entire discussion section was substantially revised. Please consider the entire new section for your review.

Line 253 “This demonstrates that physical activity is beneficial for pain control…” No, the study does not include a measure of physical activity, so it can´t be demonstrated.

Line 265 “This demonstrated that fatigue, and its components, tend to be relevant symptoms and that they deteriorate the quality of life of FM patients, regardless of their age.” No, the data showed that fatigue scores are similar between different age ranges. There is not a measure of other symptoms of FM or quality of life, so it can’t be demonstrated by the present results.

Line 268 “This may reveal that fatigue is the most persistent symptom and that there is no slowing down over the years in FM patients.” There is not a measure of other symptoms of FM for the comparison with fatigue, so it can’t be revealed by the present results.

Line 295 “The present study reinforces the real importance of assessing fatigue in patients with FM, as it is an extremely relevant symptom”. The present results showed a significant difference in fatigue scores between Portuguese and Brazilian FM patients, that’s all. Such statement can’t be declared on the basis of the present results.

It is important to be cautious and modest when interpreting the results. Even more when the data collected and the procedure are easy and direct, like in the present study. I suggest to improve the statistical analyses, think and pay attention to the interpretation of the results and rewrite with caution the discussion and conclusions. Additionally, it would be great to check the English language in the whole manuscript and complete the introduction with more bibliographic resources supporting your objective and hypothesis. Results also could be completed with the description of the obtained results.

R: We thank the reviewer for all comments and suggestions. As you will see, substantial revisions were made in the entire manuscript.